# In Vivo Performances, Carcass Traits, and Meat Quality of Pigs Fed Olive Cake Processing Waste

**DOI:** 10.3390/ani9121155

**Published:** 2019-12-17

**Authors:** Luigi Liotta, Vincenzo Chiofalo, Vittorio Lo Presti, Biagina Chiofalo

**Affiliations:** 1Department of Veterinary Sciences, University of Messina, Polo Universitario Annunziata, 98168 Messina, Italy; vittorio.lopresti@unime.it (V.L.P.); biagina.chiofalo@unime.it (B.C.); 2Meat and Agrifood Research Consortium, Polo Universitario Annunziata, 98168 Messina, Italy; 3Department of Chemical, Biological, Pharmaceutical and Environmental Sciences, University of Messina, Viale Ferdinando Stagno d’Alcontres, 31-98166 Messina, Italy

**Keywords:** finishing pig, olive cake, growth performance, meat quality

## Abstract

**Simple Summary:**

In the recent years, the use of agro-industrial by-products in animal feed has been considered to reduce costs coming from the disposing of processing wastes and from the feeding for the animal breeding. The aim of this study was to assess the inclusion of two different levels of olive cake in 72 Pietrain pigs, during the finishing period; animals fed three dietary treatments contained increasing levels of olive cake: 0% (Ctrl), 5% (Low), and 10% (High) as partial substitution of wheat middling and soybean oil. Our hypothesis was that the inclusion of olive cake could be a possible strategy for the following: (i) finding unconventional ingredients of a commercial complete feed, and (ii) improving in vivo and postmortem performances, with particular attention on the acidic profile. The present study demonstrated that feeding olive cake improved animal performances and influenced some qualitative parameters, reducing the deposition of intramuscular fat and modifying the fatty acid composition in the intramuscular fat and backfat, where the concentration of MUFA and PUFA were increased and the quality indices (AI and IT) were improved. Obviously, the inclusion of unconventional ingredients in animal feed should not negatively affect the meat organoleptic characteristics.

**Abstract:**

The aim of the study was to assess the inclusion of different levels of olive cake in pigs’ diet as a strategy to replace conventional ingredients and to improve meat quality traits. Seventy-two Pietrain pigs, during the growing–finishing period (50–120 kg BW), were fed with three dietary treatments that contained or did not contain olive cake: 0% (Ctrl), 5% (Low), and 10% (High). The trial lasted 90 days. Weekly, individual body weight (BW) and feed intake (FI) were recorded to calculate average daily gain (ADG) and feed conversion ratio (FCR). At slaughter, the dressing percentage was calculated and carcass weight and backfat thickness were measured. On a section of *Longissimus thoracis* muscle (LT), pH, color, chemical, and fatty acid composition were determined. Fatty acid profile was also determined in backfat. The statistical model included the effects of diet (Ctrl, Low, and High). The inclusion of 5% of olive cake in the diet improved significantly (*p* < 0.05) BW and FCR. Both levels of inclusion (5% and 10%) significantly reduced (*p* < 0.05) backfat thickness and intramuscular fat and modified their fatty acid composition, increasing (*p* < 0.05) the concentration of MUFA and PUFA and improving (*p* < 0.05) quality indices. Results suggest that olive cake did not negatively affect the productive performances.

## 1. Introduction

In recent years, the use of agro-industrial by-products in animal feed has been considered in order to reduce the costs for the animal breeding and the costs coming from the disposing of processing wastes according to the regulations [1]. Moreover, these by-products could replace conventional feeds, lowering the feed-to-food competition in animal production [2] and their content in bioactive compounds, such as some fatty acids and/or polyphenols, could act as functional ingredients, improving the qualitative characteristics of the meat [3].

In Italy, 1,700,000 hectares are destined for the cultivation of olives and about 80% on the southern national territory. In decreasing order, the most productive regions are Puglia (370,000 ha), Calabria (186,000 ha), and Sicily (160,000 ha); they represent about 60% of the national production. In the world, Italy is second only to Spain, with a production of 550,000, mainly represented by extra virgin and virgin oils [4]. From 100 kg of olives, about 35 kg of solid waste (olive oil cake) and 100 L of liquid waste (olive mill wastewaters) are produced and used in various ways [5].

The chemical characteristics of olive cake depends on the olive variety, the quantities of its components (skin, pulp, and stone), and the oil extraction process [6,7]. This by-product has been tested positively in pigs [8,9,10], cattle [11], small ruminants [12,13], broilers [14], and rabbits [15,16], thanks to the high quantity of residual oil. Nevertheless, the high lignin content related to the stone presence could limit its use in piglets, except that it is totally pitted [17]. On the other hand, the unsaturated fatty acid (UFA) content, especially in oleic acid, could represent an interesting functional ingredient in the diet able to positively change the acidic composition of the fat tissues [9]. In this view, the increasing request of modern society for healthy animal products with more unsaturated fatty acids (UFA), which may oppose coronary diseases linked to high cholesterol levels in blood and atherosclerosis [18], could bring the consumers closer to the products of animal origin [19]. The acidic profile of the diet in the feeding of monogastric represents a valid method to change the acidic profile of pig fat tissues and, therefore, the characteristics of products of animal origin for the human diet [20].

In this context, the aim of the study was to assess the inclusion of different levels of olive cake in finishing pig diets, as a strategy to replace conventional ingredients (such as wheat) and to improve meat quality traits.

## 2. Materials and Methods

### 2.1. Ethical Statement

All the procedures followed in this study were in accordance with the European guidelines for the care and use of animals in research [21]. The research received the institutional approval by the Regional Department of Agriculture, Rural Development and Mediterranean Fisheries—Sicilian Region (Dipartimento Regionale Agricoltura-Assessorato Regionale dell’Agricoltura, dello Sviluppo Rurale e della Pesca Mediterranea–Regione Siciliana), Italy, n. 94750023262, prot. 110180.

### 2.2. Animals, Facilities, and Experimental Diets

For the trial, 72 Pietrain barrows pigs bred on an authorized farm were studied; all pigs were clinically healthy. The animals, at 57 ± 1 kg body weight, were randomly allocated into 24 pens in a commercial farm, with three pigs per pen and eight replicates per treatment. Each pen was equipped with a nipple waterer and a stainless-steel feeder, and the pigs were given free access to feed and water throughout, in a temperature-controlled barn (20–22 °C). Daily, the incidence of pathological states and/or possible deaths were monitored.

Dietary treatments consisted in different quantities of totally pitted olive cake into the diet: 0 (Ctrl), 50 (Low), and 100 (High) g·kg^−1^ of feed as partial substitution of wheat middling and soybean oil.

The chemical characteristics of olive cake, totally pitted, used in the trial were determined according to AOAC methods [22], and the fatty acid methyl esters (FAME) were analyzed by GC–FID, following the procedure described below [23]. The analysis of each parameter was replicated three times for each by-product, and the results were reported as a mean of the three replications (Table 1).

During the finishing period, animals received the dietary treatments for 105 days, consisting of 15 days of adaptation to the experimental diets and 90 days of experimental diets. The proportion of ingredients of the dietary treatments is reported in Table 2.

Table 3 shows the nutritional characteristics and energy content of the diets. Samples of the dietary treatments were analyzed to determine the chemical composition [22] and the fatty acid profile (Table 4), following the procedure described below [23]. Each parameter was analyzed three times for each dietary treatment; Table 3 and Table 4 show the mean values of the three analyses.

### 2.3. Growth Performance, Carcass Measurements, and Meat Sampling

During the trial, weekly, on each animal, the body weight (BW) was recorded, and feed intake (FI) per pen was monitored, and, on the basis of the data, average daily gain (ADG) and feed conversion ratio (FCR) were calculated.

The day before slaughter, all the animals were individually weighed and transported to an internal farmhouse abattoir, where they were kept for 8 h, with full access to water but not feed, and then they were successively slaughtered according to standard commercial procedures. Then, each hot carcass was weighed, and the dressing percentage was calculated. At 45 min postmortem, the backfat thickness was measured between the 3rd 4th last ribs, using a ruler with a precision of 1 mm. Successively, a section of 500 (±25 g) of loin (*Longissimus thoracis* muscle (LT)), including the subcutaneous fat, was excised at the last rib level from each carcass [25]. Meat and fat samples were placed in individual plastic bags and vacuum-packaged at −20 °C, until subsequent analyses.

### 2.4. Physical Characteristics

The pH was measured, 24 h postmortem, using a pH meter (WTW 330/SET 1, Weilheim, Germany) equipped with a Hamilton double-pore glass piercing electrode, on LT muscle samples refrigerated at 4 °C. The color parameters, L* (lightness), a* (redness), and b* (yellowness), were measured on LT samples (each was a 2.5 cm thick slice), using a desktop photometer (Spectral scanner, DV Tecnologie d’Avanguardia, Padova, Italy) with an illuminating source D65 [26].

### 2.5. Nutritional Characteristics

Each sample of LT muscle was analyzed in triplicate, to determine the chemical composition according to AOAC methods [22] for moisture (no. 950.46 2010), crude protein (no. 981.10 1983), and ash (no. 920.153 and no. 923.03). The total lipids were extracted from both LT muscle (intramuscular fat—IMF) and backfat (BF), according to the ISTISAN method (1996/34 met. B page 41), with acid hydrolysis.

For the chromatographic analyses, on 15 mg of lipids, extracted using a solution of chloroform/methanol (2:1, *v*/*v*) from each sample of IMF and BF [27], the fatty acids methyl esters were prepared [28] by using a solution of sulfuric acid/methanol (1:9, *v*/*v*) and submitted to HRGC analysis, [23] with an FID detector (Agilent Technologies 6890 N, Palo Alto, CA, USA) equipped with an Omegawax 250 column (Supelco, Bellefonte, PA, USA; 30 m × 0.25 mm i.d., 0.25 μm film thickness). The concentration of each fatty acid was expressed as g/100 g, considering 100 g the sum of the areas of identified FAMEs. Each sample of IMF and BF of individual pig was analyzed three times by GC, and the results represent the mean values of the three analyses.

Data of fatty acids, from both IMF and BF, were used to calculate the quality indices (atherogenic and thrombogenic indices) in order to characterize the health benefits [29].

### 2.6. Statistical Analysis

Data were subjected to statistical analyses, using the GLM procedure of SAS [30].

Data were subjected to ANCOVA for the initial body weight, using the following model:Y_ij_ = μ + D_i_ + X_ij_ + ε_ij_
where Y_ij_ is observations, μ is the overall mean, D_i_ is the fixed effect of diets (i = 3), X_ij_ is the initial body weight, and ε_ij_ is the random residual.

Differences in body weight between groups (Ctrl, Low, and High) were not significant (*p* > 0.05); thus, body weight was removed from the data set, and data were rerun with ANOVA. Pen means represented the experimental unit for statistical analysis. Data were analyzed with the following model, using diet as the classification factor:Y_ijkl_ = μ + D_i_ + P_j_ + A_k_(i) + ε_ijkl_
where Y_ijkl_ is observations, μ is the overall mean, D_i_ is the fixed effect of diets (i = 3), P_j_ is the fixed effect of period (j = 5 for BW and FI; j = 4 for ADG and FCR), A_k_(i) is the random effect of animal k (k = 24) nested within diet I, and ε_ijkl_ is the random residual.

Data of carcass traits and meat quality were submitted to a mixed-model analysis [30], which included the fixed effect of the diet (Ctrl, Low, and High) and the random effect of the individual pig. Least square means (LSM) and standard error of least square means (SEM) were calculated. Comparisons between LSM were performed, using the Tukey test. Differences were considered significant at *p* < 0.05.

## 3. Results

### 3.1. Growth Performances and Carcass Quality

No change in the palatability of the feed integrated with the olive cake and no death were noted during the experimental period. The slaughter weight and ADG of the animals were influenced by the diet (*p* < 0.05), showing the maximum value in the pigs fed a diet with an inclusion of 50 g·kg^−1^ of olive cake (Table 5), while the feed intake and FCR value in the pigs of this group were the lowest, showing the best feed conversion ratio. Significant differences (*p* < 0.05) were observed in carcass weight, backfat thickness, and dressing percentage (Table 5), which tended to decrease linearly (*p* < 0.05) with the dietary inclusion of olive cake.

### 3.2. Meat and Fat Characteristics

Physical properties of LT muscle were not affected by the experimental treatments. In regard to the chemical composition, the by-product inclusion had a significant effect (*p* < 0.05) for the IMF content (*p* < 0.01). The increase inclusion of by-product into the diet promoted a progressive reduction of the IMF (Table 6).

Table 7 reports the effect of dietary treatment on the fatty acids in the IMF of pigs. The saturated fatty acid class (SFA) was lower (*p* < 0.05) in IMF from animals that received the dietary treatment without olive cake inclusion. Among the saturated fatty acids, the olive cake diet significantly decreased the level of palmitic and stearic acids (*p* < 0.05), and it significantly increased that of arachidic acid (*p* < 0.05). The monounsaturated fatty acid class (MUFA) was increased by olive cake inclusion (*p* < 0.05). Specifically, oleic acid was found at a higher concentration in the IMF of pigs fed the highest level of olive cake inclusion (*p* < 0.05). The sum of polyunsaturated fatty acids (PUFA) was increased by the dietary treatment (*p* < 0.05). Among the individual PUFA, the olive cake diet increased the concentration in muscle of 18:2n6 (*p* < 0.05). Atherogenic and thrombogenic indices were significantly (*p* < 0.05) lower, and therefore better, in the meat of the pigs fed a diet with 10% olive cake inclusion than that of the Ctrl and Low groups.

Table 8 reports the effect of the dietary treatment on the fatty acid composition of the BF. The inclusion of olive cake in the diet affected the sum of SFA, MUFA, and PUFA (*p* < 0.05). Among the individual SFA, the stearic acid decreased significantly (*p* < 0.05), and among the MUFA, the palmitoleic acid decreased significantly (*p* < 0.05), whereas the oleic acid increased significantly (*p* < 0.05) in the diet with 10% olive cake. Feeding the olive cake diet decreased the total SFA level significantly (*p* < 0.05) and increased the concentration of total MUFA and PUFA significantly (*p* < 0.05) in the BF.

Atherogenic and thrombogenic indices (Table 8) were significantly (*p* < 0.05) lower in the BF of the pigs fed with a 10% olive cake diet than that of the Ctrl and Low groups.

## 4. Discussion

From a productive point of view, the results suggest that the olive cake inclusion can reach up to 50 g·kg^−1^ of the diets formulated for finishing pigs, without prejudicing but improving in vitam performances, as testified by the increased ADG and reduced feed intake and FCR. These results could be due to the high fiber content in the diet of the High group; specifically, the ADL content was 100% higher than that of the Low group. The pigs fed diet with an inclusion of 50 g·kg^−1^ of olive cake reached a 13% of ADG higher than those fed the dietary treatment without olive cake inclusion. Data are in accordance with Joven et al. [9], who found a better growth rate and feed consumption in pigs fed diets with a 5% or 10% of olive cake inclusion than those fed a diet with a 15% of olive cake inclusion.

Moreover, olive cake inclusion affected the carcass weight, backfat thickness, and dressing percentage, which showed the minimum values in the High group. The results are in agreement with a study of Chiofalo et al. [31] on the effect of two fiber levels on some performance traits in Nero Siciliano fattening pigs. Authors concluded that the high-fiber diet, stimulating the growth of the GIT, which is eliminated during slaughter, caused a negative effect on pig performances and a negative incidence on carcass yield.

The backfat thickness measured at the third- to fourth-last ribs decreased linearly with the increasing of olive cake inclusion into the diet; the same significant trend was observed for the intramuscular fat content. This result could be due to the linear reduction in the energy level of the diet correlated to the linear increase of fibrous fraction content. The excessive development of part of the digestive tract in the fattening period, induced by a high fiber level, could explain this difference in carcass yield according to Garcià-Casco et al. [10]. Another explanation could be the strong positive relationships between palmitic and stearic fatty acids in backfat adipose tissue and backfat thickness. Wood et al. [20] suggest a higher dilution of exogenous polyunsaturated fatty acids with endogenous de novo synthesized fatty acids.

In regard to the chemical composition, the inclusion of olive cake negatively affected the IMF, showing a significant decrease in relation to the increasing olive cake level. This could be explained by a lower lipogenic capacity in intramuscular adipose tissue, as suggested for backfat adipose tissue [32].

The modification of meat fatty acids, through the increase of UFA and the reduction of SFA, is a strategy to follow in terms of human health. Unlike the poligastric species, the dietary unsaturated fatty acids are not desaturated in the GIT of monogastrics, and they are absorbed and accumulated unchanged in the tissues [33]. This confirms our observations on MUFA content of IMF and BF, which reflects the acidic composition of the diet, testifying the importance of the animal feeding for obtaining healthy products [34]. Furthermore, our data could also be related to the fatness percentage, according to De Smet et al. [35], who found, in monogastric and poligastric species, a high correlation between the increase of intramuscular SFA and MUFA content and the increase of fatness. This could be due to the composition of TAGs in IMF, where the presence of SFA and MUFA is higher than that of PUFA. Therefore, the increase in fatness reduces the amount of PUFA, whereas the decrease in fatness increases the amount of PUFA. According to this study, in our trial, the reduction of fatness results in an increase of intramuscular PUFA in the pigs receiving 5% and 10% of olive cake, also considering the lower PUFA content in the diets containing olive cake than that of the control diet.

Feeding olive cake significantly increased the level of oleic acid in the IMF and BF, as observed by Fontanillas et al. [36]. The sum of MUFAs in BF of pigs fed olive oil was significantly higher compared to the control diet, as observed by Scheeder et al. [37].

The PUFA content in any tissue depends on the amount and structure of dietary fat, de novo synthesis of fatty acids, the conversion rate to other fatty acids and metabolites, and the proportion of oxidation for energy consumption. In our study, there was a negative relation between the PUFA level into the diet and that into the IMF and BF, and also between the IMF content and its PUFA level, [38] and, finally, among BF thickness, fat content, and C18:2n6 level [39]. The literature [40] reports that a dietary inclusion of olive cake that is rich in MUFA reduces the polyunsaturated fatty acid content in the membranes of red blood cells. Besides, the oleic acid in pig feeding reduced the SFA and PUFA levels in raw and cooked meat [41]. Moreover, in castrated Pietrain x German Landrace pigs fed a diet with an inclusion of 5% olive oil, during the growing–finishing period, a reduction of IMF and BF polyunsaturated fatty acids was observed [42].

In the modern society, there is an increasing demand for healthy food. Clinical studies [43,44,45] report that the Mediterranean diet, characterized by a low SFA and a high MUFA consumption, especially of oleic acid coming from olive oil, which decreases the risk of heart disease and breast cancer, increasing the life expectancy. Moreover, even if the cholesterol-lowering response to MUFA is smaller than that to PUFA, their stability to lipoperoxidation, with production of metabolites characterized by adverse health effects, is higher than that of PUFA.

A recent approach suggests that AI and TI, strictly related to the saturated and unsaturated fatty acid profiles, might better characterize the health benefits of a vegetable or animal food than the n3/n6 PUFA ratio [29]. Both the indices of the intramuscular fat meat and backfat were affected by the 10% of olive cake inclusion in the pig diet. On the whole, the high content of essential MUFA and PUFA and the good value of the quality indices (AI and TI) make the lipid fraction of the pigs fed with olive cake underline the higher quality of this product for human nutrition.

The acidic profile is also responsible for the color of fat, influencing one of the most important sensorial trait of the carcass [46]. Particularly, trials [47] carried out on the measurement of meat color, using instrumental techniques, showed an influence of the polyunsaturated fatty acid content on the yellow color of fat [48]. Therefore, the fat color can be modified through the manipulation of the acidic composition of diet in the monogastric species. For this reason, the color parameters of fat, determined instrumentally, can be considered tracers of the pig feed [49]. As the color of fatty acids is neutral, the effect related to the coloring of fats may probably be due to the presence of carotenoids into the diets, responsible for the yellow color, which promote the accumulation of PUFA [48]. Papadopoulos et al. [50] in feeding laying hens with different dietary levels of an unsaturated or saturated fat source observed a significant effect on the carotenoid expression in the egg yolk at the end of experimental period.

Nevertheless, pigs fed a diet rich in carotenoids do not show accumulation of these molecules in the fat, probably because of their low level of absorption [47], so the modifications of color parameters in fat could be related to the influence of the acidic profile on other physical properties of fat, i.e., firmness, which affect the light-reflection properties [48]. In the present study, the slight differences in yellowness descriptor observed among the treatments could be due to the high polyunsaturated fatty acid levels in IMF of pigs fed high levels of olive cake.

## 5. Conclusions

The data showed that the partial substitution of wheat middling and soybean oil with olive cake affected in vivo and postmortem performances, suggesting the possibility of this by-product inclusion in diets for pigs, with a consequent reduction of costs and problems linked to the disposal of wastes. Obviously, the inclusion of unconventional ingredients in animal feed should not negatively affect the productive and sensorial performances.

In this trial, some in vivo performances and meat quality traits were positively influenced by the olive cake inclusion. Particularly, it was observed in pigs that received diets with a low or high olive cake inclusion, a better ADG and FCR, a reduction of the fat content, an increase of the MUFA and PUFA levels, and an improvement of the quality indices (AI and IT) of IMF and BF. No sensorial analysis was carried out on meat in this study, which could give additional and important information on meat quality.

The use of this low-cost by-product as feed could represent an alternative use to a feedstock for composting or direct field application, or, in the best of cases, as a fuel for biomass to energy plants, also considering that, according to the principles of Bioeconomy European Commission, the use as fertilizer or energy is a less-valued end destination of this agricultural biomass than animal feed or human food.

In this context, it is necessary to pay particular attention to the stakeholders using ad hoc formulations and methodology in order to exploit olive cake by-product in the perspective of a new feed industry able to promote rural areas and the quality of animal products.

## Figures and Tables

**Table 1 animals-09-01155-t001:** Chemical and fatty acid composition of the olive cake used in the trial.

Dried Olive Cake	As Fed
Chemical composition (g·kg^−1^)	
Moisture	48.6
Crude protein	78.6
Ether extract	277.2
Neutral detergent fiber	413.3
Acid detergent fiber	325.3
Acid detergent lignin	156.8
Crude ash	42.5
Fatty acids (g·100 g^−1^ of FAME) ^#^	
C14:0	1.76
C16:0	14.43
C18:0	3.52
C18:1n9	67.18
C18:2n6	8.39
C18:3n3	0.52

^#^ The concentration of fatty acid was expressed as g/100 g, considering 100 g the sum of the areas of all FAME identified.

**Table 2 animals-09-01155-t002:** Ingredient composition (g·kg^−1^ as fed) of the diets.

Ingredient	Ctrl	Low	High
Cornmeal	355	387.5	355
Barley	280	270	273
Soybean meal (48% CP)	130	150	135
Wheat middling	120	30	30
Field bean	72	78	75
Olive cake	0	50	100
Calcium carbonate	16	16	16
Soybean oil	13.5	5.0	2.5
Sodium chloride	3	3	3
Vit. and mineral premix ^#^	10	10	10
DL Methionine	0.5	0.5	0.5

CP = Crude Protein. ^#^ Pelleted complete feed provided the following per kg: 9000U vitamin A, 2000U Vitamin D_3_, 1.5 mg B_1_ 4 mg vitamin B_2_, 3 mg vitamin B_6_, 20l g vitamin B_12_, 30 mg vitamin E, 2.1 mg vitamin K_3_, 22.5 mg pantothenic acid, 25 mg niacin, 0.3 mg folic acid, 0.3 mg biotin, 50 mg Mn, 113 mg Zn, 125 mg Fe, 17.5 mg Cu, 1.75 mg J, 0.375 mg. Diet: Ctrl, no inclusion of olive cake; Low, inclusion of 50 g·kg^−1^ of olive cake; High, inclusion of 100 g·kg^−1^ of olive cake.

**Table 3 animals-09-01155-t003:** Nutritional characteristics (g·kg^−1^ as fed) and energy content of the diets.

Diet	Ctrl	Low	High
Chemical composition			
Moisture	99.4	97.4	98.7
Starch	447.5	443.5	436.3
Crude protein	157.0	156.7	154.1
Ether extract	40.0	42.3	46.5
Neutral detergent fiber	141.1	142.1	165.1
Acid detergent fiber	56.7	68.0	86.7
Acid detergent lignin	7.9	14.3	29.3
Crude ash	47.6	47.4	49.7
Calculated nutrient composition ^¥^			
DE (MJ·kg^−1^)	13.52	13.41	13.31
NE (MJ·kg^−1^)	9.61	9.58	9.48
Protein/energy ratio	1.16	1.17	1.16
Ca	7.6	7.5	7.3
P (total)	4.3	3.7	3.8
Na	2.9	2.9	3.0
K	7.6	7.0	7.2
Calculated total amino acid content ^¥^			
Lysine	9.4	9.3	9.1
Methionine	5.2	5.0	4.8
Tryptophan	1.7	1.7	1.5
Threonine	5.7	5.6	5.5

^¥^ According to the National Research Council (NRC) [24]. Diet: Ctrl, no inclusion of olive cake; Low, inclusion of 50 g·kg^−1^ of olive cake; High, inclusion of 100 g·kg^−1^ of olive cake.

**Table 4 animals-09-01155-t004:** Fatty acids composition (g·100 g^−1^ FAME) ^#^ of the diets.

Diet	Ctrl	Low	High
C14:0	0.23	0.21	0.19
C16:0	18.21	17.17	14.56
C18:0	3.01	2.56	2.36
C18:1n9	23.08	42.58	45.85
C18:2n6	51.75	35.70	35.06
C18:3n3	3.72	1.78	1.98
SFA	21.45	19.94	17.11
MUFA	23.08	42.58	45.85
PUFA	55.47	37.48	36.24
UFA/SFA	3.66	4.01	4.84

^#^ The concentration of fatty acid was expressed as g/100 g, considering 100 g the sum of the areas of all FAME identified. Diet: Ctrl, no inclusion of olive cake; Low, inclusion of 50 g·kg^−1^ of olive cake; High, inclusion of 100 g·kg^−1^ of olive cake. SFA = saturated fatty acids; MUFA = monounsaturated fatty acids; PUFA = polyunsaturated fatty acids. UFA/SFA = unsaturated fatty acids/saturated fatty acids ratio.

**Table 5 animals-09-01155-t005:** Effect of dietary inclusion of olive cake level on in vitam and postmortem performances of pigs.

Groups	Ctrl	Low	High	SEM	*p*-Value
Initial body weight (kg)	58.38	56.44	56.47	0.356	0.132
Final body weight (kg)	112.20 ^b^	117.19 ^a^	113.08 ^b^	0.009	0.035
Average daily gain (kg·day^−1^)	0.599 ^b^	0.675 ^a^	0.629 ^a^	0.010	0.003
Feed Intake (kg·day^−1^)	2.22 ^b^	2.20 ^b^	2.60 ^a^	0.035	0.029
Feed conversion ratio (kg·kg^−1^)	3.70 ^b^	3.28 ^b^	4.13 ^a^	0.081	0.015
Carcass weight (kg)	91.85 ^a^	90.10 ^a^	87.60 ^b^	0.013	0.033
Dressing percentage (%)	81.25 ^b^	76.88 ^a^	77.47 ^a^	0.011	0.021
Backfat thickness (3rd to 4th last ribs, mm)	40.7 ^a^	31.3 ^b^	28.0 ^b^	0.009	0.001

^a,b^ Mean values with different letter in superscript within rows indicate significant differences (*p* < 0.05). SEM, standard error of least square means. Number of replicates per treatment: *n* = 8. Diet: Ctrl, no inclusion of olive cake; Low, inclusion of 50 g·kg^−1^ of olive cake; High, inclusion of 100 g·kg^−1^ of olive cake.

**Table 6 animals-09-01155-t006:** Effect of dietary inclusion of olive cake level on physicochemical properties of *Longissimus thoracis* muscle of pigs (g·100 g^−1^).

Item	Ctrl	Low	High	SEM	*p*-Value
pH_24_	5.98	5.66	5.72	0.209	0.062
Lightness (L*)	57.32	57.13	57.87	0.720	0.455
Redness (a*)	7.05	7.32	7.34	0.263	0.376
Yellowness (b*)	10.09	9.97	10.58	0.320	0.414
Moisture	75.05	73.98	74.56	0.930	0.181
Crude protein	23.87	23.76	23.27	0.280	0.078
Intramuscular fat	2.26 ^a^	1.57 ^b^	1.33 ^b^	0.09	0.003
Crude ash	1.23	1.35	1.40	0.290	0.075

^a,b^ Mean values with different letter in superscript within rows indicate significant differences (*p* < 0.01). SEM, standard error of least square means. Number of replicates per treatment: *n* = 8. Diet: Ctrl, no inclusion of olive cake; Low, inclusion of 50 g·kg^−1^ of olive cake; High, inclusion of 100 g·kg^−1^ of olive cake.

**Table 7 animals-09-01155-t007:** Effect of dietary inclusion of olive cake level on fatty acid profile (g·100 g^−1^ FAME) ^#^ of the intramuscular fat of pigs.

Item	Ctrl	Low	High	SEM	*p*-Value
C14:0	1.14	1.33	1.55	0.320	0.169
C16:0	25.46 ^a^	25.16 ^a^	22.29 ^b^	0.181	0.006
C16:1n7	3.94 ^a^	3.57 ^a^	3.22 ^b^	0.263	0.014
C17:0	0.20	0.36	0.30	0.551	0.346
C17:1	0.20	0.22	0.20	0.297	0.132
C18:0	12.41 ^a^	11.49 ^a^	9.86 ^b^	0.856	0.022
C18:1n9	41.98 ^b^	42.41 ^b^	46.62 ^a^	0.934	0.004
C18:2n6	10.75 ^b^	11.47 ^a^	11.70 ^a^	0.345	0.028
C18.3n3	1.15	1.13	1.17	0.289	0.407
C20:0	0.28 ^b^	0.37 ^b^	0.62 ^a^	0.029	0.006
C20:1n9	0.72	0.74	0.75	0.147	0.150
C20:2	0.65	0.61	0.58	0.135	0.174
C20:3n6	0.12	0.10	0.09	0.111	0.472
C20:4n6	0.46	0.50	0.51	0.142	0.172
C20:5n3	0.03	0.04	0.04	0.108	0.373
C24:1	0.19	0.22	0.25	0.280	0.461
C22:5n3	0.17	0.15	0.13	0.175	0.098
C22:6n3	0.15	0.13	0.12	0.142	0.291
SFA	39.49 ^a^	38.71 ^a^	34.62 ^b^	1.081	0.031
MUFA	47.03 ^b^	47.16 ^b^	51.04 ^a^	1.324	0.013
PUFA	13.48 ^b^	14.13 ^a^	14.34 ^a^	0.976	0.030
AI	0.496 ^a^	0.497 ^a^	0.436 ^b^	0.028	0.022
TI	1.14 ^a^	1.105 ^a^	0.924 ^b^	0.267	0.046

^#^ The concentration of fatty acids was expressed as g·100 g^−1^, considering 100 g the sum of the areas of all identified FAMEs. ^a,b^ Mean values with different letter in superscript within rows indicate significant differences (*p* < 0.05). SEM, standard error of least square means. Number of replicates per treatment: *n* = 8. Diet: Ctrl, no inclusion of olive cake; Low, inclusion of 50 g·kg^−1^ of olive cake; High, inclusion of 100 g·kg^−1^ of olive cake. SFA = saturated fatty acids; MUFA = monounsaturated fatty acids; PUFA = polyunsaturated fatty acids. AI = atherogenic index; TI = thrombogenic index. AI = [C12:0 + (4 × C14:0) + C16:0]/[n − 6PUFA + n − 3PUFA + MUFA]. TI = [C14:0 + C16:0 + C18:0]/[(0.5 × MUFA) + (0.5 × n6PUFA) + (3 × n3PUFA) + (n3PUFA/n6PUFA)].

**Table 8 animals-09-01155-t008:** Effect of dietary inclusion of olive cake level on fatty acid profile (g·100 g^−1^ FAME) ^#^ of the backfat of pigs.

Item	Ctrl	Low	High	SEM	*p*-Value
C14:0	1.21	1.22	1.27	0.043	0.317
C16:0	24.84	23.60	22.51	0.980	0.344
C16:1	2.21 ^a^	2.14 ^a^	1.59 ^b^	0.134	0.045
C17:0	0.29	0.41	0.39	0.876	0.397
C17:1	0.29	0.28	0.26	0.09	0.108
C18:0	15.17 ^a^	14.72 ^a^	12.62 ^b^	0.867	0.032
C18:1	39.31 ^b^	38.66 ^b^	40.94 ^a^	1.876	0.036
C18:2	14.21	15.54	16.85	1.098	0.166
C18.3n3	0.81	0.79	0.78	0.354	0.264
C20:0	0.28 ^b^	0.37 ^b^	0.62 ^a^	0.029	0.004
C20:1n9	1.05	1.07	1.00	0.423	0.146
C20:2	0.67	0.65	0.54	0.175	0.091
C20:3n6	0.08	0.07	0.09	0.100	0.277
C20:4n6	0.31	0.29	0.32	0.137	0.451
C20:5n3	0.01	0.02	0.02	0.120	0.073
C24:1	0.09	0.10	0.10	0.164	0.722
C22:5n3	0.08	0.06	0.09	0.132	0.258
C22:6n3	0.03	0.01	0.01	0.143	0.749
SFA	41.79 ^a^	40.32 ^a^	37.41 ^b^	0.987	0.037
MUFA	42.01 ^b^	42.25 ^b^	43.89 ^a^	1.433	0.039
PUFA	16.20 ^b^	17.43 ^a^	18.70 ^a^	1.112	0.021
AI	0.502 ^a^	0.477 ^a^	0.441 ^b^	0.032	0.041
TI	1.290 ^a^	1.232 ^a^	1.083 ^b^	0.234	0.030

^#^ The concentration of fatty acids was expressed as g·100 g^−1^, considering 100 g the sum of the areas of all identified FAMEs. ^a,b^ Mean values with different letter in superscript within rows indicate significant differences (*p* < 0.05). SEM, standard error of least square means. Number of replicates per treatment: *n* = 8. Diet: Ctrl, no inclusion of olive cake; Low, inclusion of 50 g·kg^−1^ of olive cake; High, inclusion of 100 g·kg^−1^ of olive cake. SFA = saturated fatty acids; MUFA = monounsaturated fatty acids; PUFA = polyunsaturated fatty acids. AI = atherogenic index; TI = thrombogenic index. AI = [C12:0 + (4 × C14:0) + C16:0]/[n − 6PUFA + n − 3PUFA + MUFA]. TI = [C14:0 + C16:0 + C18:0]/[(0.5 × MUFA) + (0.5 × n6PUFA) + (3 × n3PUFA) + (n3PUFA/n6PUFA)].

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
