# Peer review of "In Vivo Performances, Carcass Traits, and Meat Quality of Pigs Fed Olive Cake Processing Waste"

_animals, 2019, doi:10.3390/ani9121155_

Round 1

Reviewer 1 Report

General remarks:

Unify the titer value in text, e.g. it is g / kg, better g · kg-1 as in Tables 1 and 2.

Provide specific data for p value in Tables 4, 5, 6, 7.

Calculate and comment on the h / H ratio.

Detailed comments are included in the text of the reviewed work

Author Response

The responses to the reviewer were indicated in red.

General remarks:

Unify the titer value in text, e.g. it is g / kg, better g · kg-1 as in Tables 1 and 2. YES, DONE

Provide specific data for p value in Tables 4, 5, 6, 7. YES, we did.

Calculate and comment on the h / H ratio.

We did not calculate this ratio. We used the Atherogenic and Thrombogenic Indices as Quality Indices in order to characterize the health benefits. A recent approach suggests that AI and TI, strictly related to the saturated and unsaturated fatty acid profiles, might better characterize the health benefits of a vegetable or animal food than the n3/n6 PUFA ratio (Ulbricht and Southgate, 1991). We reported the formulae to calculate these ratios:

AI = [C12:0 + (4×C14:0) + C16:0] / [n−6PUFA+n−3PUFA+MUFA]

TI = [C14:0 + C16:0 + C18:0] / [(0.5 × MUFA) + (0.5 × n6PUFA) + (3 × n3PUFA) + (n3PUFA / n6PUFA)]

Line 32 We added Feed intake instead of feed consumption

Table 1

Line 97: we changed acidic composition in fatty acid

We added Crude Ash

We added C18:1n9; C18:2n6; C18:3n3

Table 3

We added Crude Ash

We reported DE in MJ/kg

We added the net energy (MJ/kg) because it is more reliable than the metabolizable energy

Protein/Energy ratio. We calculated the Protein/Energy ratio.

Phosphorus Total or availability. We reported the Total Phosphorus.

How was the identical content of lysine obtained without its addition to the mixture (diets)? Sorry, there was a mistake in the table about the reported content of lysine. Moreover, the lysine content was not analyzed but it was estimated by the formulator software.

Table 4

We reported data in g · 100 g-1and we corrected data (now sum = 100)

Lines 135: All the animals were slaughtered.

Line 164 Calculate and comment on the h / H ratio.

We did not calculate this ratio. We used the Atherogenic and Thrombogenic Indices as Quality Indices in order to characterize the health benefits. A recent approach suggests that AI and TI, strictly related to the saturated and unsaturated fatty acid profiles, might better characterize the health benefits of a vegetable or animal food than the n3/n6 PUFA ratio (Ulbricht and Southgate, 1991). We reported the formulae to calculate these ratios.

Line 192: We harmonized 50 g · kg-1

Table 4 named Table 5:

We added the P values

We added the Feed intake values

Table 5 named Table 6:

We changed groups in Item

We changed pH in pH24

We changed Ash in Crude Ash

Table 6 named Table 7:

We changed groups in Item

We deleted IMF

We did not calculate h/H ratio.

Table 7 named Table 8

We changed groups in Item

We deleted BF

Line 300: We changed meat acidic profile in meat fatty acids

Line 338: We did not calculate h/H ratio

Reviewer 2 Report

The topic is interesting for readers, however I found several mistakes in experimental part of this study. Initial weight of pigs is quite high and in ctrl group weight is highest that why the carcass weight of control pigs is also highest. Backfat thickness is lowered in experimental groups but what about fat quality? We are receiving information only concerning color, no information about potential increase of fat fluidity. Diet composition in each group is a bit different (Table 2). When you enriched diet with one ingredient is better to replace also one component.

Title of Table 1 is wrong. Where do you have fat concentration of olive cake? Fatty acids composition is another story.

Table 3 – energy is better to express in MJ tan in kcal.

In Table 7 you have results of AI and TI without any description of these factors ( only one information in the text is that they are quality indices).

Line 60 – second oil extraction whose use is as fuel for broilers. I think you are speaking about energy source.

I am not shore that expression “content in unsaturated FA” is the best (line 65).

I dont like the title. In vitam = in vivo. How can you speak about post mortem performance. 

Author Response

The responses to the reviewer were indicated in red.

The topic is interesting for readers, however I found several mistakes in experimental part of this study.

Initial weight of pigs is quite high and in ctrl group weight is highest that why the carcass weight of control pigs is also highest. Initial body weight showed no significant difference among the groups. However, the statistical analysis showed that the carcass weight of the Low group was similar (P>0.05) to that of Ctrl group. It means that the differences in carcass weight could be due to the high–fiber diet of the High group and not to the higher initial body weight of the Ctrl group.

Backfat thickness is lowered in experimental groups but what about fat quality? We are receiving information only concerning color, no information about potential increase of fat fluidity. Diet composition in each group is a bit different (Table 2). First of all, our pigs were bred for use of fresh meat and not for the processing industry (cured products). However, we evaluated the consistency of back fat in relation to its fatty acid composition, especially linoleic acid which is widely used to define soft and hard adipose tissue. Our results (Table 8) showed no significant differences (P>0.05) in the linoleic acid content in relation to the experimental diets. This was confirmed by slaughterhouse operators that did not highlight technological problems during the carcass dissection procedures.

When you enriched diet with one ingredient is better to replace also one component. Our trial was carried out on commercial and not experimental diets. Therefore, the diets were not formulated by us but we used diets formulated by the feed industry which, in order to obtain isoprotein and isoenergetic diets (as you can see in Table 3), did not make a total substitution of an ingredient.

Title of Table 1 is wrong. Where do you have fat concentration of olive cake? Fatty acids composition is another story. We reported the fat concentration in olive cake as Ether Extract (277.2 g · kg-1 as fed) because of the analytical method used to determine the fat concentration.

Table 3 – energy is better to express in MJ tan in kcal. We changed the energy content in MJ.

In Table 7 you have results of AI and TI without any description of these factors ( only one information in the text is that they are quality indices). We reported in tables 7 and 8 the formulae to calculate these ratios:

AI = [C12:0 + (4×C14:0) + C16:0] / [n−6PUFA+n−3PUFA+MUFA]

TI = [C14:0 + C16:0 + C18:0] / [(0.5 × MUFA) + (0.5 × n6PUFA) + (3 × n3PUFA) + (n3PUFA / n6PUFA)].

We used the Atherogenic and Thrombogenic Indices as Quality Indices in order to characterize the health benefits. A recent approach suggests that AI and TI, strictly related to the saturated and unsaturated fatty acid profiles, might better characterize the health benefits of a vegetable or animal food than the n3/n6 PUFA ratio (Ulbricht and Southgate, 1991).

Line 60 – second oil extraction whose use is as fuel for broilers. I think you are speaking about energy source. We deleted this sentence to avoid confusion.

I am not shore that expression “content in unsaturated FA” is the best (line 62). We changed in unsaturated fatty acid (UFA) content.

I dont like the title. In vitam = in vivo. How can you speak about post mortem performance. 

We Changed the title in “In vivo performances, carcass traits and meat quality of pigs fed olive cake processing waste”.

Reviewer 3 Report

General Comments

The current manuscript presents research findings on performance and carcass quality of finishing pigs fed diets supplemented with olive cake. The current findings could be used as a basis for further field experiments in order to discover proper inclusion levels of similar products. The authors do not describe adequately the limitations of the study in several parts of the manuscript, and therefore major revision is required (abstract, discussion, and conclusions, see specific comments). Also, due to the fact that reformulation of the diets took place, authors should revise the specific parts in the introduction and material and method section. As it is existing form, there is very limited information on the nutritional strategy followed. Moreover, statistics should be rerun in the sense that initial body weight in the beginning of fattening to be used as covariate, as growth of pigs is positively correlated with the body weight in each phase.

Specific Comments

Lines 13-15: sentence is long and there is repetition twice of the word costs. Needs to be revised.

Lines 16-18: should contain information what was the concept of reformulation of the diets in incorporating olive cake at these levels, in other words what was removed from the ration. Where the new diets more economic compared to the control? Such information is missing here.

Line 30: starting and ending body weight should be provided, as these may differ from country to country and also between farms

Lines 74-75: in terms of ingredients to be replaced wheat is referred here, but in the rations used in the study wheat middling was the major ingredient replaced, and partially a reduction of soybean oil. The authors should have mentioned in the introduction part, which possible ingredients could be replaced by the olive cake in the diets of growing pigs. Based on its composition, it terms of starch, protein, fat, fiber content what alternatives could be proposed. This is a point to be addressed and negotiated by the authors thoroughly.

Line 164: in the statistical evaluation of the data, apart from the random effect of the animal, the body weight at the beginning of the experiments should have been tested as a covariate in the model. If not significant then it should be removed and rerun the statistics. The individual body weight may have contributed to the growth potential of the animals as such a correlation has been documented for pigs. It is recommended to the authors to perform this procedure and correct accordingly the results.

Line 188: the heading of the Table should define the categories-treatments of the animals (ctrl, low, high) so the reader will not look back in the text

Line 188, Table 4: feed intake is not provided in the Table. Please proceed with the necessary addition

Lines 258-260: the term higher level of inclusion should be revised, as in the present study only 2 levels were tested and a tolerance dosage was not used in the treatments provided.

Lines 269-270: could it be attributed also to the fact that IMF was increased, so it may be suggested that due to the different nature of fatty acids, oleic acid enriched diets tend to “accumulate” PUFA intramuscularily.

Line 289-290: please clarify this point better. The increase in PUFA is related to the reduction of total fat, but not to the diet FA profile?

Line 297: How these correlations were demonstrated? Statistically?

Line 319-320: why the n3/n6 ratio was not calculated?

Lines 335-337: In laying hens it was recently demonstrated that feeding them with variable fatty acid profile and unsaturated to saturated fatty acid ratio, on the long term influences the carotenoid concentration in the egg yolk. It is plausible an explanation for the observations here.

Line 340: wheat was not substituted, it was middlings. Also soybean oil was reduced.

Author Response

The responses to the reviewer were indicated in red.

General Comments

The current manuscript presents research findings on performance and carcass quality of finishing pigs fed diets supplemented with olive cake. The current findings could be used as a basis for further field experiments in order to discover proper inclusion levels of similar products. The authors do not describe adequately the limitations of the study in several parts of the manuscript, and therefore major revision is required (abstract, discussion, and conclusions, see specific comments). Also, due to the fact that reformulation of the diets took place, authors should revise the specific parts in the introduction and material and method section. As it is existing form, there is very limited information on the nutritional strategy followed. Moreover, statistics should be rerun in the sense that initial body weight in the beginning of fattening to be used as covariate, as growth of pigs is positively correlated with the body weight in each phase. We rerun the statistical analysis (see below).

Specific Comments

Lines 13-15: sentence is long and there is repetition twice of the word costs. Needs to be revised. OK, we rephrased this sentence.

Lines 16-18: should contain information what was the concept of reformulation of the diets in incorporating olive cake at these levels, in other words what was removed from the ration. We added as partial substitution of wheat middling and soybean oil.

Where the new diets more economic compared to the control? Such information is missing here.

The cost of olive cake was 0.21 €/kg. The costs of the Low and High diets were the same of the Ctrl diet (0.31 €/kg) but, in the view of disposing of processing wastes, according to the European regulations, and improving meat quality, the substitution appeared interesting.

Line 30: starting and ending body weight should be provided, as these may differ from country to country and also between farms. Added information (50-120 kg of BW).

Lines 70-72: in terms of ingredients to be replaced wheat is referred here, but in the rations used in the study wheat middling was the major ingredient replaced, and partially a reduction of soybean oil. The authors should have mentioned in the introduction part, which possible ingredients could be replaced by the olive cake in the diets of growing pigs. Based on its composition, it terms of starch, protein, fat, fiber content what alternatives could be proposed. This is a point to be addressed and negotiated by the authors thoroughly. We spoke with the formulator of the feed industry; usually the use of olive cake is an alternative to cereal by-products (fibers) and oils (energy).

Line 168: in the statistical evaluation of the data, apart from the random effect of the animal, the body weight at the beginning of the experiments should have been tested as a covariate in the model. If not significant then it should be removed and rerun the statistics. The individual body weight may have contributed to the growth potential of the animals as such a correlation has been documented for pigs. It is recommended to the authors to perform this procedure and correct accordingly the results. Yes, at the beginning, we tested body weight as a covariate and since it was not significant (P>0.05), body weight was removed and the data, together with the in vivo parameters (ADG, FI, FCR), rerun to ANOVA as reported in the manuscript.

Line 203 – 217 – 250 - 264: the heading of the Table should define the categories-treatments of the animals (ctrl, low, high) so the reader will not look back in the text. OK, done. We added at the end of the Tables the legend: Diet: Ctrl, no inclusion of olive cake; Low, inclusion of 50 g · kg-1 of olive cake; High, inclusion of 100 g · kg-1 of olive cake.

Line 193, Table 4: feed intake is not provided in the Table. Please proceed with the necessary addition. Yes, we added the feed intake values.

Lines 258-260: the term higher level of inclusion should be revised, as in the present study only 2 levels were tested and a tolerance dosage was not used in the treatments provided. Yes, right. We deleted this sentence to avoid confusion.

Lines 269-270: could it be attributed also to the fact that IMF was increased, so it may be suggested that due to the different nature of fatty acids, oleic acid enriched diets tend to “accumulate” PUFA intramuscularily. No. The IMF decreased (P<0.01) linearly (NOT INCREASED) with the increasing of olive cake inclusion into the diet. 

Line 309-313: please clarify this point better. The increase in PUFA is related to the reduction of total fat, but not to the diet FA profile? We rephrased the sentence in: “Therefore, the increase in fatness reduces the amount of PUFA, whereas, the decrease in fatness increase the amount of PUFA. According to this study, in our trial, the reduction of fatness results in an increase of intramuscular PUFA in the pigs receiving 5% and 10 % of olive cake, also considering the lower PUFA content into the diets containing olive cake than that of the control diet”.

Line 319: How these correlations were demonstrated? Statistically? No, we did not. We changed the word correlations in “relation between……”

Line 333-338: why the n3/n6 ratio was not calculated? We did not calculate this ratio. We used the Atherogenic and Thrombogenic Indices as Quality Indices in order to characterize the health benefits. A recent approach suggests that AI and TI, strictly related to the saturated and unsaturated fatty acid profiles, might better characterize the health benefits of a vegetable or animal food than the n3/n6 PUFA ratio (Ulbricht and Southgate, 1991). We reported the formulae to calculate these ratios where you can see a higher inclusion of fatty acids involved (such as  C12:0, C14:0, C16:0, C18:0) which better characterize a food in relation to the cardiovascular disease:

AI = [C12:0 + (4×C14:0) + C16:0] / [n−6PUFA+n−3PUFA+MUFA]

TI = [C14:0 + C16:0 + C18:0] / [(0.5 × MUFA) + (0.5 × n6PUFA) + (3 × n3PUFA) + (n3PUFA / n6PUFA)]

Lines 347-349: In laying hens it was recently demonstrated that feeding them with variable fatty acid profile and unsaturated to saturated fatty acid ratio, on the long term influences the carotenoid concentration in the egg yolk. It is plausible an explanation for the observations here. Yes, we added this study in the Discussion: “Papadopoulos et al. (2019) in feeding laying hens with different dietary levels of an unsaturated or saturated fat source observed a significant effect on the carotenoid expression in the egg yolk at the end of experimental period.”

Line 359: wheat was not substituted, it was middlings. Also soybean oil was reduced. Yes, we changed the phrase in “Data showed that the partial substitution of wheat middling and soybean oil with olive cake…..”

Round 2

Reviewer 3 Report

The authors have complied with all the recommended corrections and additions. The manuscript has been improved after all adjustments were implemented.

No further comments are addressed.

Author Response

Thank you for your reply.

Sincerely

Luigi Liotta

This manuscript is a resubmission of an earlier submission. The following is a list of the peer review reports and author responses from that submission.